# Encoding Spatial Distribution of Convolutional Features for Texture Representation

**Yong Xu**[1,2], **Feng Li**[1], **Zhile Chen**[1], **Jinxiu Liang**[1], **Yuhui Quan**[1*]

[1] School of Computer Science and Engineering, South China University of Technology, China
[2] Peng Cheng Laboratory, China

`yxu@scut.edu.cn, csfengli@mail.scut.edu.cn, cszhilechen@mail.scut.edu.cn,`
`cssherryliang@mail.scut.edu.cn, csyhquan@scut.edu.cn`

## Abstract

Existing convolutional neural networks (CNNs) often use global average pooling (GAP) to aggregate feature maps into a single representation. However, GAP cannot well characterize complex distributive patterns of spatial features while such patterns play an important role in texture-oriented applications, *e.g.*, material recognition and ground terrain classification. In the context of texture representation, this paper addressed the issue by proposing Fractal Encoding (FE), a feature encoding module grounded by multi-fractal geometry. Considering a CNN feature map as a union of level sets of points lying in the 2D space, FE characterizes their spatial layout via a local-global hierarchical fractal analysis which examines the multi-scale power behavior on each level set. This enables a CNN to encode the regularity on the spatial arrangement of image features, leading to a robust yet discriminative spectrum descriptor. In addition, FE has trainable parameters for data adaptivity and can be easily incorporated into existing CNNs for end-to-end training. We applied FE to ResNet-based texture classification and retrieval, and demonstrated its effectiveness on several benchmark datasets.

## 1 Introduction

Ideal image descriptors for classification are capable of characterizing both local features and global patterns of an image. Local features are about the local singularities or fine details of an image, such as edges and corners. Global patterns are about the distributions and spatial layouts of local image features. While local features are critical for visual recognition, global image patterns also play an important role in many scenarios, such as material recognition [1], ground terrain classification [2] and dynamics recognition [3].

Typical convolutional neural networks (CNNs) for image classification connect convolutional layers with fully-connected (FC) layers. The FC layers encode the spatial layouts of feature maps and act as a classifier. Such a structure fails to produce a robust global representation, as FC layers capture spatial layouts based on "absolute" locations and their outputs are not invariant to spatial transforms such as translation, rotation and scaling, that often occur in real scenarios.

To achieve the robustness to spatial transforms in CNN-based representation, many existing studies (*e.g.* [4, 5, 6]) apply global average pooling (GAP) to feature maps, which averages all entries in a feature channel so as to eliminate the dependency on spatial locations. However, the naive averaging operation totally disregards the spatial layout of a feature map. As a result, when dealing with images of complex distributive patterns, GAP cannot encode sufficiently discriminative information.

---

*Corresponding author: Yuhui Quan

35th Conference on Neural Information Processing Systems (NeurIPS 2021).

A robust yet discriminative global representation is particularly important for analyzing images of texture patterns [7], since for such images, spatial transforms are one main source of variations and distributive characteristics are the key clues during analysis. In the context of texture analysis and related applications, there have been some efforts [8, 9, 10, 11, 2, 12] on designing a more sophisticated global pooling module. Most of these works are about encoding the numerical distribution of features (*e.g.* soft histograms [8, 9, 12]) into a single representation. Nonetheless, they may still lose the details on the spatially distributive characteristics of features which are useful for texture analysis.

Aiming at characterizing the spatially distributive patterns in feature maps to form a robust yet discriminative texture representation, this paper proposed a statistical global feature encoding module called FE (Fractal Encoding), which is grounded by fractal geometry [13, 14].

**Motivations**    Fractal geometry has been widely used for texture analysis and synthesis; see *e.g.* [15, 16, 17, 18, 19, 20]. Its central concept is the fractal dimension that measures the distributive irregularity of a spatial point set. Many studies have shown that natural textures carry fractal dimension information [15, 16, 21]. Since a CNN is a spatial feature extractor, its generated feature maps may also encode fractal dimension information that reflects the regularity of texture patterns. Inspired by this, FE describes the spatial layout of features in terms of spatially distributive regularity using fractal-dimension-based statistics in a hierarchical manner.

Fractal dimension is established based on the measurement at scale $\delta$. For each $\delta$, an object is measured in a way that ignores the irregularity of size less than $\delta$. If the measurement is proportional to $\delta^{-\beta}$ for some $\beta$ and $\beta$ is almost the same for small scales $\delta$, we can compute its limit, which is called the fractal dimension. Let the $n$-dimensional Euclidean space $\mathbb{R}^n$ be covered by a mesh of $n$-dimensional hypercubes with diameter $d$. Given a point set $\mathbb{E} \subset \mathbb{R}^n$, its fractal dimension denoted by $\beta(\mathbb{E})$ is defined by [14]:

$$\beta(\mathbb{E}) = \lim_{d \to 0} \frac{\log \mathcal{N}(\mathbb{E}, d)}{-\log d}, \tag{1}$$

where $\mathcal{N}(\mathbb{E}, d)$ is the number of mesh hypercubes that intersect $\mathbb{E}$.

It can be seen fractal dimension is about cross-scale examination in terms of power law. This makes it possible to encode additional discriminative information on spatial layouts that ignored by the simple number counting operations in a histogram. For simplicity, consider the fractal dimensions on binary images. The white/black pixels on a binary image can be viewed as a set of points on a 2D grid. See Figure 1 for three numerical examples, where the fractal dimensions are different between the grid and the two bars, while the histograms of all the sample images in terms of black/white pixels are the same. If we view those sample images as specific feature patches, the fractal-dimension-based statistics from FE can well distinguish the spatial feature organization of those two types of textures.

In addition to discriminability, fractal dimension is theoretically invariant to spatial bi-Lipschitz transform [21, 20], a general transform that covers a wide range of often-seen image transformations. Note that the influence of the spatial transforms of an image are likely to propagate to its feature maps. Then the FE driven by fractal dimensions can enjoy the robustness in this case.



Figure 1: Fractal dimensions (numbers in the second row) calculated by the scheme of [22] on a grid image and two bar images. The white pixels are viewed as a set of spatial points on a 2D grid. If we view the binary images as some texture features (*e.g.* feature patches from a CNN), then fractal dimensions can well distinguish the spatial distribution of the two types of texture features.

**Main idea**    The key part in FE is the fractal analysis pooling (FAP). For higher discriminability, FAP leverages multi-fractal geometry [23] to form multiple fractal-dimension-based statistical quantities. This leads to a local-global hierarchical fractal analysis process. Concretely, FE looks at a feature map as an union of point sets. The point sets are the level sets under a local point-wise categorization function $D(\cdot)$. The FE calculates a single (global) fractal-dimension-based statistical quantity on

every level set:

$$f(\widehat{D}) = \beta(\mathbb{E}_{\widehat{D}}), \text{ where } \mathbb{E}_{\widehat{D}} = \{\boldsymbol{x} \in \mathbb{E} : D(\boldsymbol{x}) = \widehat{D}\}. \tag{2}$$

Encouraged by the robustness of fractal dimension to spatial transforms and feature value scaling, we introduce the function $D$ based on local fractal dimension [16] (also called Hölder exponent [14]) defined as the power $\mathcal{D}(\cdot)$ of $\mathcal{U}\left(\mathcal{B}\left(\boldsymbol{x}, r\right)\right) \propto r^{\mathcal{D}(\boldsymbol{x})}$ for $\boldsymbol{x} \in \mathbb{E}$, where $\mathcal{B}\left(\boldsymbol{x}, r\right)$ is a hypercube with center $\boldsymbol{x}$ and diameter $2r$, and $\mathcal{U}$ is some measurement. It can be seen that the local fractal dimension has a similar form as the global one in (1):

$$\mathcal{D}(\boldsymbol{x}) = \lim_{r \to 0} \frac{\log \mathcal{U}(B(\boldsymbol{x}, r))}{-\log r}. \tag{3}$$

In other words, $\mathcal{D}(\boldsymbol{x})$ characterizes the local power law around $\boldsymbol{x}$ under measurement $\mathcal{U}$.

The above local-global hierarchical fractal analysis process involves some discrete operations (*e.g.* box counting) that are non-differentiable, which creates obstacles for back propagation and model training. By appropriate design with soft relaxations, we implement FE using common operations of CNNs, so that it can be end-to-end trained on top of any standard CNN and deal with images of varying sizes. In addition, FE is designed with trainable parameters which can be jointly learned with the backbone network for better adaptivity to data, as well as for automatically setting some hyper-parameters which may be difficult to determine but critical to the performance in practice.

Since FAP and GAP capture distributive features from different aspects respectively, FE combines them with the bi-linear pooling (BLP) [24] to exploit their complementary capabilities. The resulting module is deployed to the ResNet for texture classification and retrieval.

**Contributions**    This work contributes to deep learning and texture analysis in the following aspects. Firstly, we propose a novel fractal-analysis-driven global feature encoding module which is capable of encoding the spatially distributive regularity of features on a feature map. Secondly, we build multi-fractal analysis into a standard CNN for boosting the power of deep-learning-based texture representation, which leads to state-of-the-art results in classification and retrieval tasks.

## 2  Related Work

**Global pooling for deep texture representation**    The GAP is a prominent choice for global feature aggregation; see *e.g.* [4, 5] for its applications in texture representation. Squeezing a feature map by averaging, GAP is very lossy regarding spatial information. To address this, most existing studies drew inspirations from traditional global descriptors. Cimpoi *et al.* [25] and Song *et al.* [26] used fisher vector encoding [27]. but the resulting network is difficult for end-to-end training. Similar to kernel-based descriptors [28], the BLP proposed by Lin and Maji [24] is based on the outer product of each pair of feature points, which encodes more information than GAP in terms of dependencies among channels. Dai *et al.* [29] combined BLP and GAP. However, both BLP and GAP are not trainable. Following bag-of-features [30], Zhang *et al.* [8] proposed a trainable texture encoding (TE) layer which captures the numerical distribution of features based on dictionary learning. Xue *et al.* [9] combined TE and GAP via BLP. A similar module is used in Hu *et al.* [10], but it is placed on top of multiple convolutional layers besides the last one, with an FC layer for result ensemble. Motivated by locality-constrained coding [31], Bu *et al.* [11] proposed a locality-aware coding layer which encodes features with considerations on their locality constraints. To make histograms applicable to CNNs, Peeples [12] proposed a histogram layer, which is used together with GAP. Chen *et al.* [32] exploited the soft histogram of cross-layer local fractal dimensions for feature encoding.

The proposed FE in this work distinguishes itself from the aforementioned methods by encoding the regularity on the spatial distribution of convolutional features, providing additional discrimination.

**Deep CNNs for texture classification**    In addition to global pooling, there are also some studies focusing on other aspects of deep-learning-based texture classification. To utilize the inherent correlations among the spatial contexts corresponding to visual attributes of texture images, Zhai *et al.* [33] proposed a multiple-attribute-perceived network with a multi-branch architecture. To efficiently describe the spatial organization of complex texture, Zhai *et al.* [34] proposed to learn the dependencies among texture features with a self-attention-like module, with very impressive results achieved.

**Fractal geometry for texture representation**    Fractal analysis has a long history for texture representation. Previous methods mostly employ fractal dimensions (implemented with various numerical

calculation schemes) as local appearance descriptors [16] or global distribution descriptors [15, 21]. These fractal-based descriptors are handcrafted and not adaptive to data. In comparison, our work forms a deep-learning-based fractal analysis model which can be end-to-end trained from data for better adaptivity.

# 3 Fractal Encoding: Pooling with Hierarchical Fractal Geometry

The proposed FE module processes a feature tensor from the CNN backbone with two paths that perform GAP and FAP receptively. An FC layer is attached behind GAP to align GAP to FAP in terms of output size, magnitude and semantics. An upsampling layer is applied before FAP for higher spatial resolutions of feature maps so as to make FAP more stable. The results from the two paths are integrated by BLP followed by an FC layer. See Figure 2 for the pipeline of the FE and FAP.

The FAP first compresses the input feature tensor to $C_0$ channels by a $\text{Conv}_{1\times1}\to\text{BN}\to\text{ReLU}$ block. On the new feature tensor, FAP processes it slice by slice and then concatenates the results over all slices. By viewing the entries on a feature slice as a set of points in $\mathbb{R}^2$, FAP runs a hierarchical fractal dimension analysis on it, *i.e.*, FAP groups the points based on local fractal dimension and calculates a fractal-dimension-based statistical quantity on each group. Accordingly, there are three main blocks in FAP, as described in the next subsections.

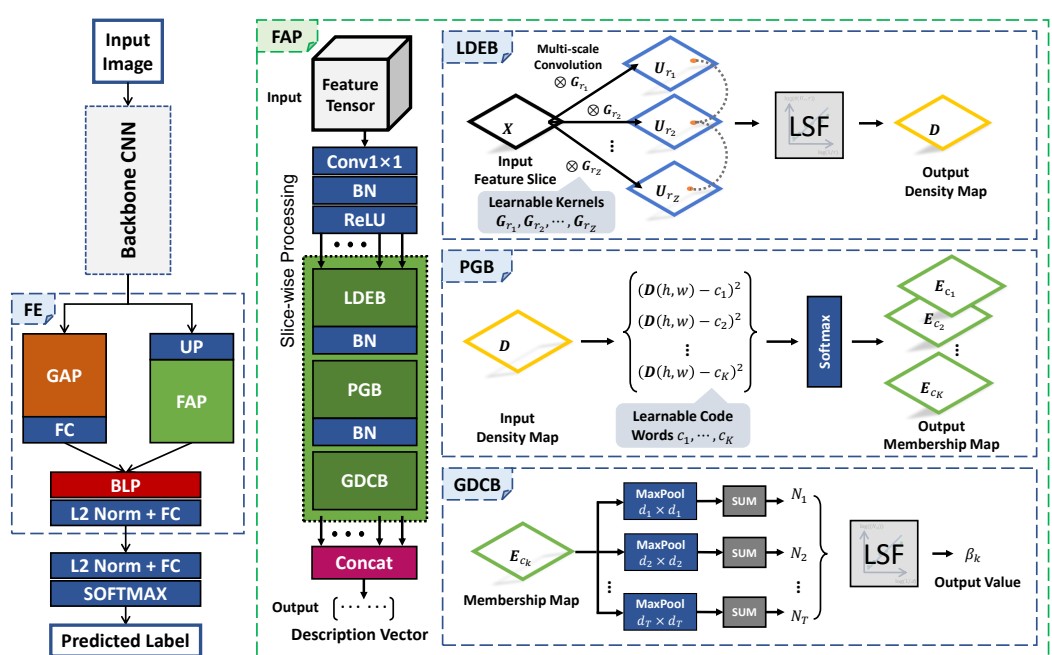

Figure 2: Diagram of proposed fractal encoding (FE) module.

## 3.1 Local Dimension Estimation Block (LDEB)

The LDEB calculates a point-wise local fractal dimension map $\boldsymbol{D} \in \mathbb{R}^{H\times W}$ from the input feature slice $\boldsymbol{X} \in \mathbb{R}^{H\times W}$:

$$\text{LDEB} : \boldsymbol{X} \in \mathbb{R}^{H\times W} \to \boldsymbol{D} \in \mathbb{R}^{H\times W}. \tag{4}$$

The calculation is based on (3), where one key is computing the measurement $\mathcal{U}$, *i.e.*, forming a series of measurement maps: $\boldsymbol{U}_r \in \mathbb{R}^{H\times W}, r = 1, 2, \cdots$.

The measurement $\mathcal{U}$ in (3) calculates a value on local block $\mathcal{B}(\boldsymbol{x}, r)$ for every $\boldsymbol{x}$. For a feature slice, a local block can be viewed as a local patch of size $r \times r$ in the slice, and $\mathcal{U}$ is essentially calculated on all the patches sampled with a sliding window on $\boldsymbol{X}$, which can be modeled by a convolution process. We define the measurement to be learnable for better data adaptivity. Since the measurement is defined on multiple scales, we simulate this by a multi-scale convolution:

$$\boldsymbol{U}_r = \boldsymbol{G}_r \otimes \boldsymbol{X}, \;\; r = 1, 2, \cdots, \tag{5}$$

where $\otimes$ denotes the discrete convolution operation, and $\boldsymbol{G}_r \in \mathbb{R}^{r \times r}$ is an $r \times r$ kernel learned together with the model.

The numerical implementation of (3) is done by the least-squares fitting (LSF) on $\boldsymbol{U}_r$ versus $r$ in the log-log coordinate system. A series of measurement maps $\{\boldsymbol{U}_r, r = r_1, r_2, \cdots, r_Z\}$ are first calculated, where $r_1 < \cdots < r_Z$ are a finite sequence of predefined ordered positive radii. Then the estimation of local fractal dimension map in (3) becomes an LSF problem:

$$\min_{\boldsymbol{D}, \boldsymbol{P}} \sum_{z=1}^{Z} (\boldsymbol{D}(w,h) \log r_z + \boldsymbol{P}(w,h) - \log \boldsymbol{U}_{r_z}(w,h))^2, \tag{6}$$

where $\boldsymbol{P}(w,h)$ denotes the bias term in the fitting. Setting the derivative of (6) *w.r.t.* $\boldsymbol{D}$ and $\boldsymbol{P}$ to zero respectively, we have

$$\begin{cases} \sum_{z=1}^{Z} \left( \boldsymbol{D}(w,h) \log^2 r_z + \boldsymbol{P}(w,h) \log r_z - \log r_z \log \boldsymbol{U}_{r_z}(w,h) \right) = 0 \\ \sum_{z=1}^{Z} \left( \boldsymbol{D}(w,h) \log r_z + \boldsymbol{P}(w,h) - \log \boldsymbol{U}_{r_z}(w,h) \right) = 0 \end{cases} \tag{7}$$

Then, the local fractal dimension map $\boldsymbol{D} \in \mathbb{R}^{W \times H}$ is computed as the solution to (7):

$$\boldsymbol{D}(w,h) = \frac{Z \sum_{z=1}^{Z} \log r_z \log \boldsymbol{U}_{r_z}(w,h) - \left( \sum_{z=1}^{Z} \log r_z \right) \left( \sum_{z=1}^{Z} \log \boldsymbol{U}_{r_z}(w,h) \right)}{Z \sum_{z=1}^{Z} \log^2 r_z - \left( \sum_{z=1}^{Z} \log r_z \right)^2}. \tag{8}$$

## 3.2 Point Grouping Block (PGB)

The PGB categorizes the entries on the feature slice based on the map $\boldsymbol{D}$ output by the LDEB. Let

$$\mathbb{E} = \{1, \cdots, W\} \times \{1, \cdots, H\} \subset \mathbb{R}^2, \tag{9}$$

$$\mathbb{E}_{\widehat{D}_k} = \{(h, w) \in \mathbb{E} : \widehat{D}_k \leq \boldsymbol{D}(h, w) < \widehat{D}_{k+1}\}, \tag{10}$$

for $k = 1, \cdots, K$, given the intervals $[\widehat{D}_1, \widehat{D}_2], [\widehat{D}_2, \widehat{D}_3], \cdots, [\widehat{D}_K, \widehat{D}_{K+1}]$ where $\widehat{D}_k \in \mathbb{R}$ for all $k$. The set $\mathbb{E}_{\widehat{D}_k}$ can be represented by a binary map $\boldsymbol{E}_{\widehat{D}_k}$ of the same size as $\boldsymbol{D}$, where $\boldsymbol{E}_{\widehat{D}_k}(h, w) = 1$ if $\widehat{D}_k \leq \boldsymbol{D}(h, w) < \widehat{D}_{k+1}$ and 0 otherwise.

The setting of intervals $[\widehat{D}_k, \widehat{D}_{k+1}], k = 1, \cdots, K$ can have a noticeable impact on the accuracy of multi-fractal analysis [23]. However, they are non-trivial to determine, as the range of $\boldsymbol{D}$ is uncertain and uniform intervals are often not optimal. In addition, grouping with hard assignment may cause noticeable quantization errors [35], as well as optimization difficulty in back propagation during model training. To address these issues, we implement the grouping function using a soft assignment scheme [35] with trainable quantization intervals. Instead of using binary values, the soft assignment uses real values in $[0, 1]$ for defining $\boldsymbol{E}_{c_k}(h, w)$ to indicate the degree of membership. Let $\{c_k \in \mathbb{R}, \ k = 1, ..., K\}$ denote a set of trainable interval anchors. For each $c_k$, we define its corresponding membership map $\boldsymbol{E}_{c_k} \in [0, 1]^{W \times H}$ by

$$\boldsymbol{E}_{c_k}(w, h) = \frac{\exp(-s_k(\boldsymbol{D}(w, h) - c_k)^2)}{\sum_{k=1}^{K} \exp(-s_k(\boldsymbol{D}(w, h) - c_k)^2)}, \tag{11}$$

where $\{s_k \in \mathbb{R}\}_{k=1}^{K}$ is a set of learnable weights. It can be implemented via Softmax. To summarize, the PGB takes a local dimension map $\boldsymbol{D}$ as input, and generates a series of soft membership maps:

$$\text{PGB} : \boldsymbol{D} \in \mathbb{R}^{H \times W} \rightarrow \{\boldsymbol{E}_{c_k} \in [0, 1]^{H \times W}\}_{k=1}^{K}. \tag{12}$$

## 3.3 Global Dimension Calculation Block (GDCB)

The GDCB accepts $\{\boldsymbol{E}_{c_k}\}_{k=1}^{K}$ from the PGB as input, calculates a fractal-dimension-based quantity $\beta$ on each $\boldsymbol{E}_{c_k}$ individually, and finally concatenates $\beta_k$s as output:

$$\text{GDCB} : \{\boldsymbol{E}_{c_k} \in [0, 1]^{H \times W}\}_{k=1}^{K} \rightarrow \boldsymbol{\beta} = (\beta_1, \beta_2, \cdots, \beta_K) \in \mathbb{R}^K. \tag{13}$$

The quantity $\beta_k$ is calculated based on (1). Similar to the computation on $\boldsymbol{D}$, Equation (1) can be numerically implemented by using bi-log LSF. However, the box counting operation in (1) is about

counting the patches with non-zero entries, which is non-differentiable and thus makes training infeasible. Also recall that $\boldsymbol{E}_{c_k}$ is a soft membership map with values in $[0, 1]$ instead of binary ones. It is very likely that $\boldsymbol{E}_{c_k}$ contains no zeros and the box counting is inapplicable in this case.

Based on above considerations, we replace the box counting $\mathcal{N}$ in (1) by the sum of the maximal values from all patches of size $d \times d$ in $\boldsymbol{E}_{c_k}$. When $\boldsymbol{E}_{c_k}$ is binary, it equals to box counting. The new scheme can be efficiently implemented by stride-1 max pooling with window size $d \times d$ and summation. By varying $d = d_1, \cdots, d_T$, we have the sums denoted by $N_1, \cdots, N_T$. Similar to (7), $\beta_k$ is then calculated as the LSF solution to the modified (1):

$$
\beta_k = \frac{T \sum_{t=1}^{T} \log d_t \log N_t - \left( \sum_{t=1}^{T} \log d_t \right) \left( \sum_{t=1}^{T} \log N_t \right)}{T \sum_{t=1}^{T} \log^2 d_t - \left( \sum_{t=1}^{T} \log d_t \right)^2}. \tag{14}
$$

## 4 Experimental Evaluation

### 4.1 Deployment of FE Module and Setting of Experiments

The FE module can be easily incorporated into a CNN by porting it on top of some convolutional layers. To evaluate the effectiveness of FE, we insert it next to the last convolutional layer of the ResNet [36], an often-used backbone (*e.g.* [8, 9]). The resulting CNN is referred to as FENet. Inheriting the advantages from FE, FENet can deal with flexible image size and be end-to-end trained.

We apply FENet to texture classification and texture retrieval on six benchmark datasets, including GTOS [2], GTOS-M [9], KTH [37], MINC [38], DTD [39] and FMD [1]. Same as existing work, we use the provided split schemes on GTOS, MINC and DTD, and random 10 splits on KTH-TIPS2b and FMD with recommended split sizes. The mean and standard deviation of classification accuracies over all splits are calculated. The results are reported using two runs on GTOS-M and five-time statistics on other datasets.

Based on the empirical parameter setting in fractal analysis in previous studies [21, 35], we set $C_0 = 3$ in GAP, $r = 1, 2, ..., 6$ in LDEB, $K = 16$ in PGB, and $d_t = 2, 3, \cdots, 6$ in GDCB. The output sizes of the two paths in FE are both set to $48$. Following [34, 8], we use ResNet-18 and ResNet-50 as the backbone respectively. On all the datasets, FENet is trained with the cross-entropy loss via the momentum SGD optimizer with default settings and 30 epochs. The batch size is set to 16 on FMD, 32 on KTH, and $64$ on the other four datasets. The learning rate is initialized to $1e^{-3}$ on FMD, $5e^{-3}$ on MINC, GTOS and GTOS-M, and $1e^{-2}$ on KTH and DTD datasets, with cosine decay every 10 epochs. The ResNet backbones are initialized with the pre-trained models on Imagenet. The $\{\boldsymbol{G}_r\}_r$ in (5) are initialized as Gaussian kernels with bandwidth 1. Other parameters of FENet are initialized by Xavier [40]. Data augmentation via horizontal flipping and random cropping to $224 \times 224$ is applied. Our FENet is implemented with PyTorch 1.7, and the code will be released at the website: `https://github.com/csfengli/FENet`. All the experiments were run on a single Titan XP GPU.

### 4.2 Results in Texture Classification

Our FENet is compared with several recent deep models for texture classification, which include DeepTEN [8], DEPNet [9], LSCNet [11], MAPNet [33], DSRNet [34] and HistNet [12]. The results on the six datasets are listed in Table 1. The results of the compared methods are directly quoted from existing literature whenever possible, or left blank otherwise.

It can be seen from Table 1 that FENet performs the best on KTH, FMD, MINC and GTOS in terms of mean accuracy, no matter using ResNet-18 backbone or ResNet-50 backbone. On GTOS-M, the FENet is the best performer with ResNet-18 backbone.

On DTD, the FENet performs better than others except DSRNet and MAPNet. The reason why FENet shows unsatisfactory performance on DTD is probably that, the labels for the textures in DTD are based on describable subjective semantics, which are quite different from other texture datasets whose labels are based on objective materials. Even in such a case, our method still applies and provides a good texture descriptor (e.g. superior performance to DeepTEN, DEPNet and HistNet). Recall that MAPNet is developed based on multiple attribute perception (MAP) which is more suitable for

Table 1: Classification accuracies (%) in the form of "mean±s.t.d.". Best results are **boldfaced**. The precision varies as the results are quoted from different works.

| | Method | DTD | KTH | FMD | MINC | GTOS | GTOS-M |
|---|---|---|---|---|---|---|---|
| **ResNet-18** | DeepTEN | - | - | - | - | - | 76.12 |
| | DEPNet | - | - | - | - | - | 82.18 |
| | LSCNet | - | - | 76.3 | - | - | - |
| | MAPNet | $69.5 \pm 0.8$ | $80.9 \pm 1.8$ | $80.8 \pm 1.0$ | - | $80.3 \pm 2.6$ | $82.98 \pm 1.6$ |
| | DSRNet | $\mathbf{71.2} \pm 0.7$ | $81.8 \pm 1.6$ | $81.3 \pm 0.8$ | - | $81.0 \pm 2.1$ | $83.65 \pm 1.5$ |
| | HistNet | - | - | - | - | - | $79.75 \pm 0.8$ |
| | Ours | $69.59 \pm 0.04$ | $\mathbf{86.62} \pm 0.08$ | $\mathbf{82.26} \pm 0.29$ | $\mathbf{80.57} \pm 0.10$ | $\mathbf{83.10} \pm 0.23$ | $\mathbf{85.10} \pm 0.36$ |
| **ResNet-50** | DeepTEN | 69.6 | $82.0 \pm 3.3$ | $80.2 \pm 0.9$ | 81.3 | $84.5 \pm 2.9$ | - |
| | DEPNet | 73.2 | - | - | 82.0 | - | - |
| | LSCNet | - | - | 81.2 | - | - | - |
| | MAPNet | $76.1 \pm 0.6$ | $84.5 \pm 1.3$ | $85.2 \pm 0.7$ | - | $84.7 \pm 2.2$ | $86.64 \pm 1.5$ |
| | DSRNet | $\mathbf{77.6} \pm 0.6$ | $85.9 \pm 1.3$ | $86.0 \pm 0.8$ | - | $85.3 \pm 2.0$ | $\mathbf{87.03} \pm 1.5$ |
| | HistNet | $72.0 \pm 1.2$ | - | - | $82.4 \pm 0.3$ | - | - |
| | Ours | $74.20 \pm 0.10$ | $\mathbf{88.24} \pm 0.23$ | $\mathbf{86.74} \pm 0.19$ | $\mathbf{83.98} \pm 0.14$ | $\mathbf{85.71} \pm 0.06$ | $85.20 \pm 0.43$ |

the subjective attributes on DTD (similar reason for DSRNet), while the spatial regularity captured by our FE is more suitable for encoding the characteristics of objective materials (*e.g.* the superior performance of our method to MAPNet and DSRNet on other datasets).

It is worth mentioning that the training dataset of GTOS-M is divided into three subsets with small, medium and large scales respectively. It is shown in [9] that the performance of deep CNNs such as DeepTEN using the single-resolution subset rather than the multi-resolution ones of GTOS-M will have a noticeable accuracy decrease. In comparison, our FENet has no performance decrease when trained on single-scale data, and the results of FENet on GTOS-M in Table 1 are actually based on only the medium-scale training data. See Section 4.5 for more analysis.

It is also mentioning that the performance improvement of DSRNet and MAPNet is less than 1% in most cases, which is so significant and seems to saturate. In comparison, in the case of ResNet-18, there are three datasets where our method has accuracy improvement larger than 1% and even over 4.5%, and on the remaining dataset the improvement is nearly 1%.

## 4.3 Results in Texture Retrieval

The global representation to the classification module in FENet can be used as a texture descriptor. Its effectiveness is evaluated by the retrieval task as follows. An image sample is represented by its corresponding feature tensor output by the last pooling layer (*i.e.* after BLP). Each sample in the test dataset is used as a query to search the remaining samples, and its top-$R$ similar samples in the dataset are collected in terms of cosine similarity. The average precision and recall over all query samples are calculated.

By varying $R$ from 1 to the number of searched samples, we obtain the precision-recall (PR) curves and the corresponding mean average precision (mAP), which are shown in Figure 3. It can be seen that FENet achieved the best retrieval results among the compared methods on GTOS-M, KTH and FMD datasets. Such results have demonstrated the effectiveness of the texture feature representation learned in FENet.

## 4.4 Effectiveness of FE Module

**Ablation study on FE module in FENet**     To demonstrate the effectiveness of FE in FENet, we form a baseline ("w/o FE Module") from FENet by removing its FE module as well as the BLP layer and adapting the dimension of the related FC layer. Similarly, we form another baseline ("w/o GAP") by removing the GAP layer in FENet. These two baselines are trained in the same way as FENet.

The results of the baselines are compared to FENet on GTOS-M and KTH datasets in Table 2. It can be seen that the performance decrease caused by the removal of FE Module is significant on GTOS-M and KTH. This suggests that the regularity of spatial distribution captured by FE Module

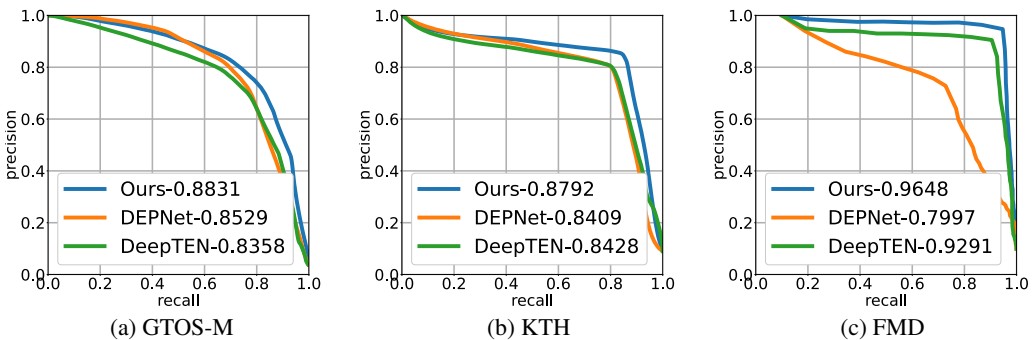

|              (a) GTOS-M               |              (b) KTH               |              (c) FMD               |

Figure 3: PR curves of several methods in texture retrieval. The values in legends are mAP.

plays a more important role. When we remove the GAP in FE, performance decreases moderately. All these results have demonstrated the effectiveness of FE Module in FENet.

Table 2: Classification accuracy (%) of FENet with ablations.

| Dataset | Backbone | w/ FE Module [FENet] | w/o FE Module | w/o GAP |
|---------|----------|---------------------|---------------|---------|
| GTOS-M | ResNet-18 | $85.10 \pm 0.36$ | $82.29 \pm 0.30$ | $83.86 \pm 0.64$ |
| KTH | ResNet-18 | $86.62 \pm 0.08$ | $83.25 \pm 0.76$ | $86.43 \pm 0.07$ |
| GTOS-M | ResNet-50 | $85.20 \pm 0.43$ | $82.57 \pm 0.65$ | $84.33 \pm 0.09$ |
| KTH | ResNet-50 | $88.24 \pm 0.23$ | $83.82 \pm 0.25$ | $87.59 \pm 0.15$ |

**Incorporating FE module into DeepTEN**    We also demonstrate the effectiveness of the FE module by plugging it into DeepTEN [8] and examining the performance gain. Recall that the DeepTEN also uses a ResNet backbone, together with an texture encoding module (called TE) which acts as a soft histogram with learned bins. Similar to our FENet, the FE module (together with the upsampling layer in front of it) is inserted in parallel to the TE layer after the last RB of the ResNet backbone. The output of the FE module and TE module are integrated by BLP. The resulting model is trained with the same scheme as the original work.

The results on the four datasets are listed in Table 3. It can be seen that the introduction of FE module brings significant improvement to DeepTEN on GTOS-M, KTH and FMD, as well as noticeable improvement on MINC. Such improvement again demonstrated the effectiveness of the FE module. Indeed, the TE module in DeepTEN is essentially a soft counting layer which only considers the quantitative distribution but ignores the spatial distribution of features in feature slices. The FE module provides complementary information to the TE module by characterizing the spatial organization of a feature map based on multi-fractal analysis.

Table 3: Classification accuracy (%) of DeepTEN w/o and w/ FE module.

| Dataset | Backbone | DeepTEN | DeepTEN + FE Module |
|---------|----------|---------|---------------------|
| GTOS-M | ResNet-18 | 76.12 | $81.15 \pm 0.29$ |
| KTH | ResNet-50 | $82.00 \pm 3.30$ | $86.93 \pm 0.30$ |
| FMD | ResNet-50 | $80.20 \pm 0.90$ | $86.32 \pm 0.43$ |
| MINC | ResNet-50 | 81.3 | $83.62 \pm 0.11$ |

### 4.5   More Analysis

**Complexity and efficiency**    We compare the complexity of backbone CNN, DeepTEN, DEPNet and FENet in terms of the number of parameters and number of floating operations per second (FLOPS). We also compare their efficiency in term of inference time on an input image of size $384 \times 384$. See Table 4 for the comparison. Our FE module only introduces a small number of

additional parameters to the backbone, and the complexity of FENet is comparable to that of other deep models under the same backbone.

Table 4: Number of parameters, number of FLOPs and inference time on GPU of different models. The values on the left/right are for the ResNet-18/ResNet-50 backbones respectively. The inference time is tested on the input of size $384 \times 384$.

| Method | # Parameters ($1 \times 10^6$) | # FLOPS ($1 \times 10^9$) | GPU Time (s/100pics) |
|---|---|---|---|
| Backbone CNN | 11.1924 / 23.5716 | 1.8185 / 4.1094 | 0.1387 / 0.3665 |
| DeepTEN | 11.3736 / 23.9017 | 1.8229 / 4.1234 | 0.1165 / 0.3000 |
| DEPNet | 12.2679 / 25.4841 | 1.8333 / 4.1627 | 0.0751 / 0.3160 |
| FENet | 11.5071 / 23.9323 | 1.8205 / 4.1159 | 0.1393 / 0.4530 |

**Generalization across scales**    Scaling is one often-seen variation of images which is challenging for texture analysis. Empirically, the FENet enjoys good generalization to scale changes. An experiment is conducted on the GTOS-M dataset whose training/test set is divided into three subsets with resolution of $256 \times 256$, $384 \times 384$ and $512 \times 512$ respectively; see Figure 4 for some examples. We train the ResNet-18 baseline, DEPNet and FENet only using the subset of resolution $384 \times 384$. The ResNet-18 is used as the backbone of DEPNet as well as FENet. Then the models are used as the global descriptors on the test set of resolution $384 \times 384$, as well as on both the training and test sets of other two resolutions. This is done by taking the features generated from the GAP layer in ResNet-18 baseline or the BLP layer in DEPNet and FENet. Figure 5 visualizes the descriptions via t-SNE [41]. It can be seen that the descriptions generated by FENet are more concentrated than those of the other two models.

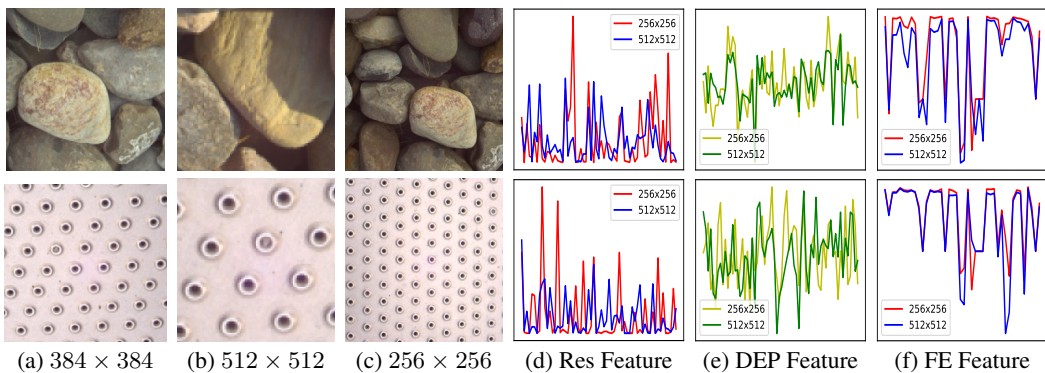

(a) $384 \times 384$    (b) $512 \times 512$    (c) $256 \times 256$    (d) Res Feature    (e) DEP Feature    (f) FE Feature

Figure 4: Multi-scale sample images in GTOS-M and their features generated by Residual, DEP and FE modules. The ResNet1-18 backbone is used. Res Feature denotes the features from the backbone.

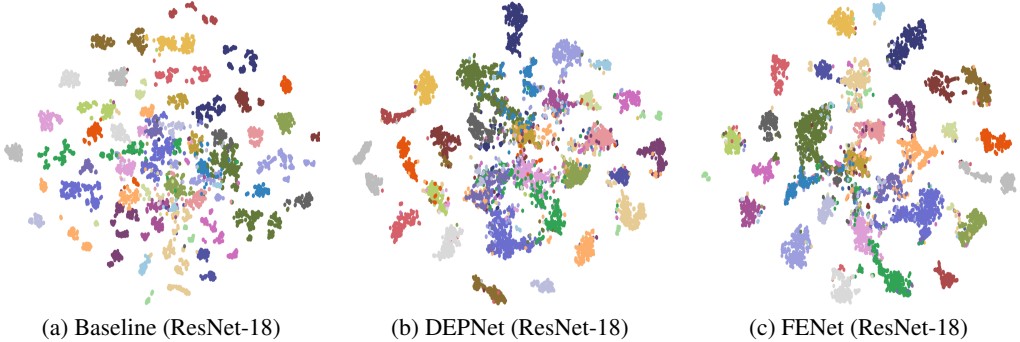

(a) Baseline (ResNet-18)    (b) DEPNet (ResNet-18)    (c) FENet (ResNet-18)

Figure 5:  The t-SNE visualization results on the scale-variant images of GTOS-M.

See also Figure 4 for the visualization on the descriptions generated on specific images with different scales. The FENet outputs descriptions with higher degrees of invariance to scale changes. Such a

robustness is probably due to the robustness of the fractal-dimension-based features to bi-Lipschitz transform as well as the cross-scale analysis in FE.

**Study on confusion cases**    The confusion matrices on the test set of GTOS-M are shown in Figure 6. We can see that the FENet is more effective than the ResNet backbone and DEPNet, with higher accuracy in classifying shale, stone cement and plastic cover. See Figure 7(a) for some sample images that confuse the ResNet backbone but correctly classified by FENet. The samples at the top row are misclassified into the class of the corresponding samples at the bottom rows. Each of such sample pairs share similar textons but with discrepancy on the distribution of textons. By leveraging fractal geometry analysis in FE, the FENet can utilize such discrepancy for discrimination, while the ResNet backbone cannot. However, from Figure 6, FENet still misclassifies sand to soil, stone asphalt to asphalt, and large limstone to cement. Some misclassified samples are shown in Figure 7(b). The sample images from the categories of 'Sand' and 'Soil' have very similar appearances and spatial distributions of patterns, which are hard to distinguish even by human. See also 'Cotton' versus 'Wood', as well as 'Polished Stone' versus 'Ceramic'.

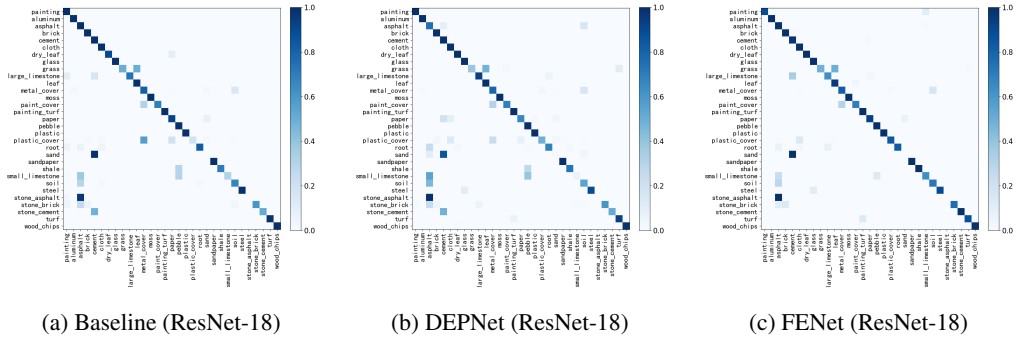

(a) Baseline (ResNet-18)        (b) DEPNet (ResNet-18)        (c) FENet (ResNet-18)

Figure 6: Confusion matrices on GTOS-M.

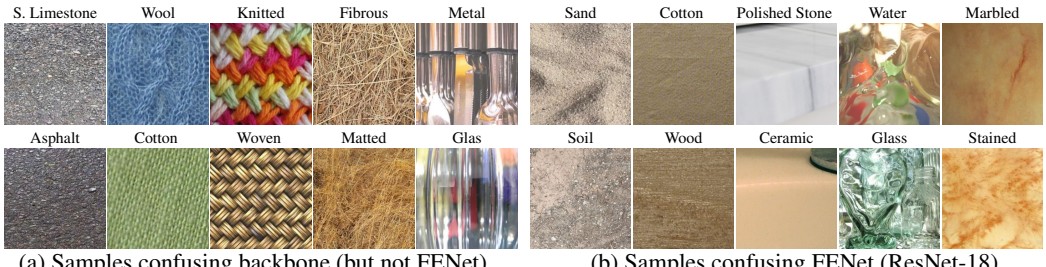

(a) Samples confusing backbone (but not FENet)        (b) Samples confusing FENet (ResNet-18)

Figure 7: Confusion analysis. The samples at the top row are incorrectly classified into the class of the corresponding samples at the bottom rows.

## 5    Summary

Feature aggregation is a key step in many CNN-based vision tasks, particularly for texture analysis. It aims at transforming feature maps into a single robust yet discriminative representation, by eliminating the dependency of features on spatial order while encoding the distributive characteristics of features. In this work, we fulfilled this goal by the proposed FE module, which leverages multi-fractal geometry and uses a hierarchical fractal analysis process for encoding the regularity of spatial arrangement in a feature map. The FE module can be easily incorporated into standard CNNs for end-to-end training. Building FE into ResNet, we developed the FENet for texture classification and retrieval. The experiments on six benchmark datasets have demonstrated the effectiveness of FENet.

While we only considered static textures in the experiments, the proposed FE can also benefit the representation and analysis on dynamic textures. We will investigate this topic in our future work. In addition to texture representation, the FE can also be incorporated into the CNNs in other visual tasks that need to utilize the spatial organization of local patterns. We also leave this to our future work.

## Acknowledgments and Disclosure of Funding

Yong Xu would like to acknowledge the support from National Natural Science Foundation of China (Grant No. 62072188, 61672241), Natural Science Foundation of Guangdong Province (Grant No. 2016A030308013), and Science and Technology Program of Guangdong Province (Grant No. 2019A050510010). Yuhui Quan would like to acknowledge the support from National Natural Science Foundation of China (Grant No. 61872151), Natural Science Foundation of Guangdong Province (Grant No. 2020A1515011128), and CCF-Tencent Open Fund 2020.

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
