# Supplementary Materials

## 1   Additional Retrieval Results

We also conduct the retrieval experiments on UIUC dataset [1], using the model trained on GTOS-M dataset. Figure 1 shows the retrieval results for a query image. The number of retrieval results in same category as the retrieved image reaches 2 out of 10 by DeepTEN18, while the FENet18 retrieves all correctly. This indicates the superior discriminability of the texture representation generated by our FENet. Indeed, the samples confuse DeepTEN have similar texture elements but distinct spatial arrangement of the elements. FENet can encode such distinct feature for an accurate retrieval.

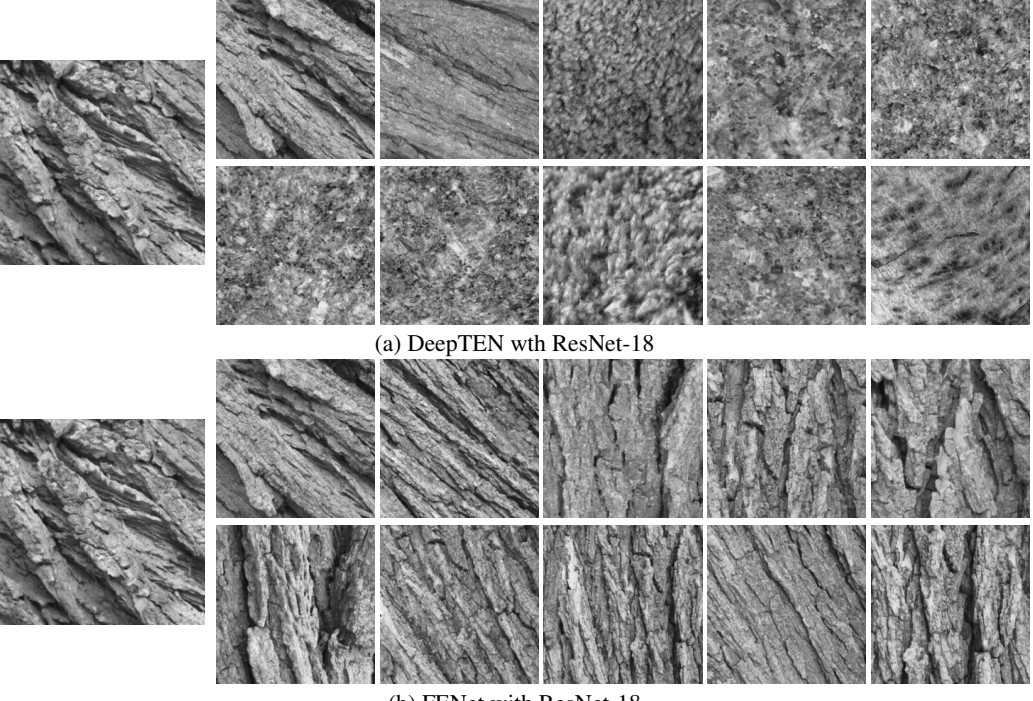

(a) DeepTEN wth ResNet-18

(b) FENet with ResNet-18

Figure 1: Retrieval results on UIUC dataset. The image shown in first column is used as the query image and the retrieved images are shown in other columns with a descend order of similarity.

35th Conference on Neural Information Processing Systems (NeurIPS 2021).

## 2 t-SNE Visualization

In Fig. 5 of main paper, we show the t-SNE visualization on the generalization of different across scales. Now we show the t-SNE visualization using the models trained with the standard training set that contains multiple scales. The t-SNE visualization is done on 10000 images from the GTOS-M dataset in terms of the feature tensor output by the BLP layer of the FENet and the output by the GAP layer of the ResNet baseline. The results are shown in Figure 2. In comparison to ResNet baseline, the texture representations generated by FENet are compacter, with larger intra-class variation and smaller inner-class variation. This demonstrates that the FENet can generate texture description with stronger robustness and higher discrimination for individual classes.

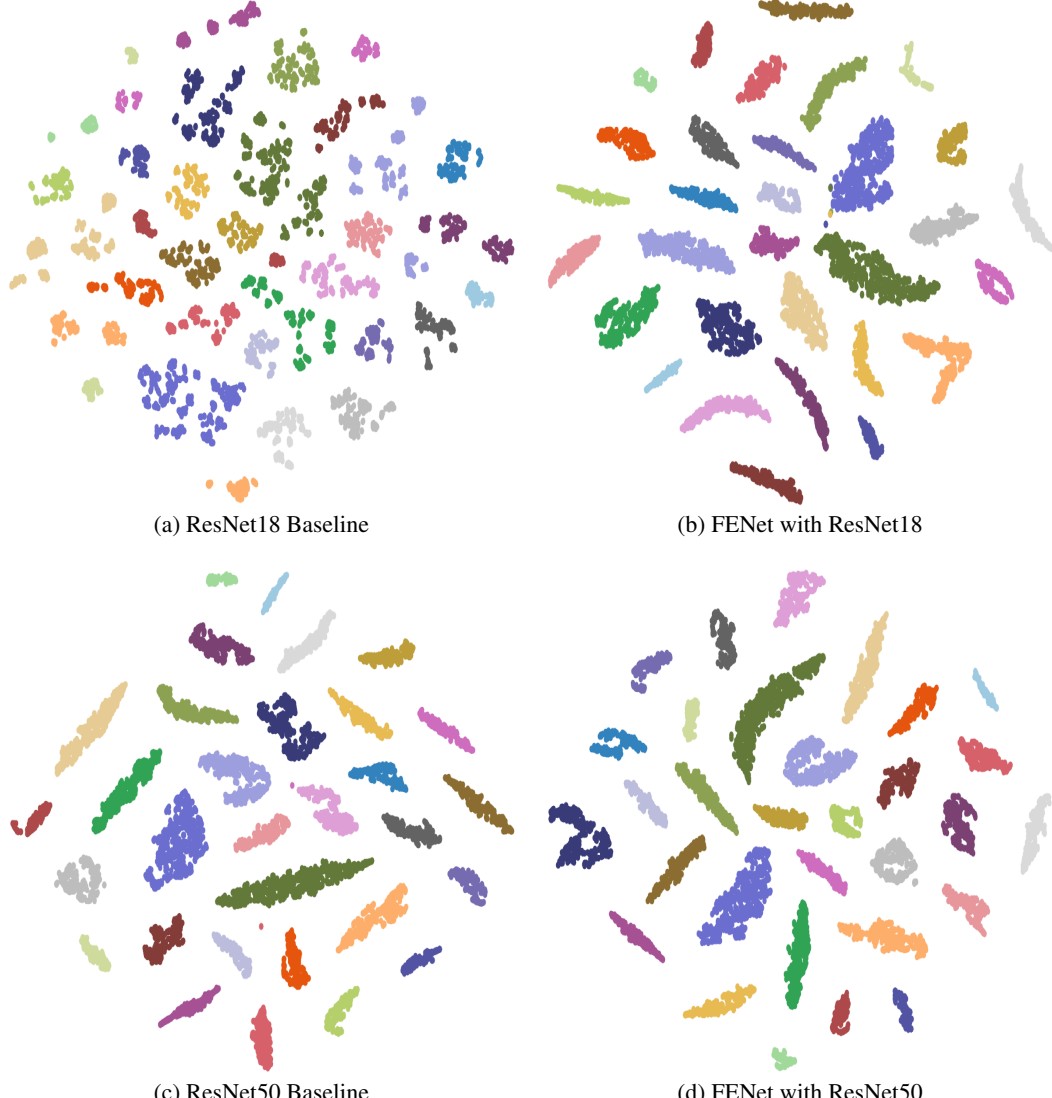

(a) ResNet18 Baseline

(b) FENet with ResNet18

(c) ResNet50 Baseline

(d) FENet with ResNet50

Figure 2: tSNE visualization on GTOS-M dataset.

## 3 Broader Impact

The results of this paper have the potential applications in texture synthesis, scene recognition, and scene parsing. For instance, the descriptor generated by FE can be used for guiding the generation of texture regions with certain spatial layout.

In addition, the results of this paper suggest that, exploiting traditional statistical approaches in image classification for designing CNN architectures may be one promising direction for deep-learning-based pattern classification. Thus our work may inspire new CNN designs along this line.

Regarding negative impacts, any image classification system that learns from data runs the risk of producing biased or misleading results reflective of the training data. Such a risk can cause serious consequences and high cost in some cases, *e.g.* incorrect medical diagnosis.

## References

[1] Svetlana Lazebnik, Cordelia Schmid, and Jean Ponce. A sparse texture representation using local affine regions. *IEEE Trans. Pattern Anal. Mach. Intell.*, 27(8):1265–1278, 2005.