# OpenReview forum: "Encoding Spatial Distribution of Convolutional Features for Texture Representation"
_NeurIPS.cc/2021/Conference — NeurIPS 2021 Poster_

### Official Review · Reviewer_H7rW · 2021-07-15

**Rating:** 6
**Confidence:** 3

**Summary:**

The authors propose complementing global average pooling in convolutional networks with another pooling method that aims at better representation of texture. This pooling method builds on fractal geometry and is implemented by a fractal encoding module. The module first locally estimates fractal dimension, the estimates are binned and inside each bin the fractal dimension estimation is applied again. This results in pooling of one input channel into a fixed-sized vector. The operations, such as binning and counting, are adapted to make the module differentiable. The experiments show that both types of pooling modules are required for good performance.

**Limitations And Societal Impact:**

Suggestions:
- Consider showing & exploring the errors/failures of the proposed method.
- Fig. 2 would benefit from a larger font ("Learnable Kernels", "Output Density Map")
- Please check if the leading term in the numerator and in denominator of (8) should be multiplied by Z.
- Explain how where the parameters of the module selected (line 196).
- Add the input image size used for Table 4.
- Fig. 4 - labels for (b) and (c) seem to be flipped, label for (d) should probably read "TE Feature".
- Fig. 4 - would it be possible to add one more panel for the baseline feature? And to all three "Feature" panels add the 384x384 series.
- Fig. 4 - please enlarge the font in the "Feature" panels.
- Fig. 5c - specify if this is FENet-18 or FENet-50.
- Fig. 4 shows DeepTEN for comparison but Fig. 5 shows DEPNet for comparison. Consider using a consistent set of baselines.

Questions:
- What are the differences between the t-SNE visualization in Fig. 5 and SFig. 4?
- Why are DeepTEN and DEPNet selected as the most common baselines when Table 1 shows that their results are relatively poor? DSRNet would seem to make more sense.
- Any comments on why is FENet clearly slower than DEPNet even if FENet requires less FLOPS than DEPNet?

Typos:
- L25: output -> outputs
- L103: point -> points / term -> terms

**Main Review:**

The proposed method is new and provides improved performance for texture classification and retrieval. The method is relatively clearly explained, but there is definitely room for improvement, see detailed suggestions and questions below.

The question is whether the FE module has any application beyond texture-related tasks as it is known that convolutional networks, even without this module, focus too much on texture cues.

**Time Spent Reviewing:**

4

---

> ### Author Response · Authors · 2021-08-10
> **Responses to the comments from Reviewer H7rW**
>
> We sincerely thank the reviewer for his/her constructive comments.
>
> ----------------------------------------------------------------------------------
> **Regarding “The question is whether the FE module has any application beyond texture-related tasks as it is known that convolutional networks, even without this module, focus too much on texture cues.“**
>
> First, we would like to note that the texture-related applications are very wide, as texture is one fundamental visual cue. While existing CNNs may focus much on texture cues, the convolutional feature maps can be viewed as spatial features which need to be aggregated into a useful statistical descriptor for achieving both robustness and discriminability. Our work provides a new CNN module that is capable of encoding the spatial distribution of convolutional features.
>
> Our paper shows that the FE module can characterize the spatial distribution, which provides additional discrimination to histogram statistics that only considers numerical distribution rather than spatial distribution. In other words, in the applications where histogram-based statistics is used for encoding the numerical distribution of spatial elements, the FE module can be used to provide a different statistical measure on spatial distributive features.
>
> ------------------------------------------------------------------------------
> **Regarding the differences between the t-SNE visualization in Fig. 5 and SFig. 4**
>
> The t-SNE visualization in Fig. 5 (main paper) is on the experiment on the generalization across scales. We train the model by using the single-scale (384$\times$384) training set and test the model with test sets of other two scales: 256$\times$256 and 512$\times$512. In SFig. 4 (supplementary materials), the model is trained with the standard training set that contains multiple scales. Comparing the t-SNE visualization results we can find that, using single-scale training makes the features more scattered but our method still enjoys higher robustness than other methods in this case.
>
> ------------------------------------------------------------------------------
> **Regarding why using DeepTEN and DEPNet rather than DSRNet as baselines in many experiments**
>
> Since the codes of DSRNet and MAPNet have not been made public available, we can only quote the quantitative results from their original work and cannot obtain other results for comparison. The DeepTEN and DEPNet are two representative approaches with published codes that can reproduce their results. Thus, we use DeepTEN and DEPNet as the baselines in other experiment parts.
>
> ------------------------------------------------------------------------------
> **Regarding why FENet runs clearly slower than DEPNet even if FENet requires less FLOPS than DEPNet**
>
> Thanks for the comment. When running on CPU with single thread, fewer FLOPs generally lead to shorter inference time. That is why FLOPs is used as a metric in our table. However, when running on GPU, there are many factors in implementation that can influence the inference time, such as parallel programming tricks, sync overhead, memory access cost, number of element-wise operations, and environment support; please see the following reference for more details:
>
> "Practical Guidelines for Efficient CNN Architecture Design, ECCV 2018."
>
> For instance, DEPNet has a similar structure to DeepTEN with more parameters and more FLOPs; however, the implementation of DEPNet favors GPU computation and thus runs much faster than DeepTEN.
>
> Our current implementation of FENet has not been fully optimized for GPU computation yet. For instance, there is still some idle time for the sync of multiple branches. We will continuously refine our GPU implementation.
>
> ------------------------------------------------------------------------------
> **Regarding the suggestions**
>
> Thanks a lot for the suggestions. Please see below for our responses.
>
> - Failure cases: We did provide such cases on which our method misclassified. Please refer to Section I in the supplementary materials. We will show more in revision.
>
> - Z in the leading terms of Equation (8): We checked it. It is necessary according to least-squares formula.
>
> - Parameter choosing at Line 196. These parameters are in the LDEB, PGB and GDCB, which are the parameters in fractal analysis. We mainly refer to previous studies on fractal analysis [14,16] to have a simple empirical setting on these parameters. We will add the explanation in revision.
>
> - Figure 2: The font size will be enlarged.
>
> - Table 4: The input image size is 384$\times$384. We will add it.
>
> - Figure 4: We will revise it as suggested. Note that such a revision does not affect the analysis and conclusion made in the paper.
>
> - DeepTEN in Figure 4. It is a typo, which should be DEPNet, as the main paper describes. Thus, it is consistent between Figure 4 and 5. We will fix the typo.
>
> - Figure 5c: It is about Resnet-18. We will add the description.

---

> > ### Comment · Reviewer_H7rW · 2021-08-17
> > **Thanks**
> >
> > Thanks for the explanations addressing my concerns.

---

### Official Review · Reviewer_wSwZ · 2021-07-16

**Rating:** 6
**Confidence:** 4

**Summary:**

The authors propose fractal encoding (FE) module to encode spatial distribution of convolutional features for texture representation. The effectiveness of FE is shown with experiments about texture classification and retrieval.

**Limitations And Societal Impact:**

yes

**Main Review:**

The proposed fractal encoding module is simple yet efficient (e.g., parameter number, Flops, and running time). It has trainable parameters for data adaptivity and also can be inserted into existing CNN structures.
The main results on texture classification and retrieval demonstrate the effectiveness of the proposed method.
The ablation study with visual results well shows the generalization across scales.
The paper writing and organization are good and easy for readers to follow.
Would the authors provide the code of the method?

After rebuttal

Thanks for providing the feedback. I also read comments from other reviewers and like to keep my original rating.

**Time Spent Reviewing:**

11

---

> ### Author Response · Authors · 2021-08-10
> **Responses to the comments from Reviewer wSwZ**
>
> We sincerely thank the reviewer for his/her constructive comments.
>
> ------------------------------------------------------------------------------
> Yes. **We promise that our code will be definitely made publicly available upon paper’s acceptance.**

---

> > ### Comment · Reviewer_wSwZ · 2021-08-23
> > **After rebuttal**
> >
> > Thanks for providing the feedback.
> >
> > I also read comments from other reviewers and would like to keep my original rating.

---

### Official Review · Reviewer_H23L · 2021-07-16

**Rating:** 6
**Confidence:** 4

**Summary:**

This paper proposes a new "global" pooling approach that aims at preserving detailed/fine-grained information from the representation. The proposed approach is based on Fractal Encoding that are implemented such that the overall representation is still trainable. The proposed approach is tested on the task texture representation.

**Ethical Concerns:**

Nothing that caught my attention

**Limitations And Societal Impact:**

While the authors provided different ablations, they did not explicitly discuss the limitations of their approach. It would be useful to dedicate a small subsection to this as this would benefit progress on this domain.
Negative societal impact is not applicable in my opinion.

**Main Review:**

Generally, the paper addresses an interesting downside of using Global Average Pooling (GAP) in CNNs, however this downside is only specific to certain situations IMHO, so I believe that the proposed solution is not broadly applicable. The following points highlight some more specific pros and cons.

Pros:
+ The paper is clear and well written.
+ The proposed solution based on Fractal Encoding is well motivated and technically sound.
+ Making sensitive hyper-parameters, such as quantization intervals, trainable is a good approach over using some ad-hoc solution.

Cons:
-  Generally the empirical results do not seem to speak strongly in favour of the proposed approach. In particular, Table 1 is either below state-of-the-art or shows that the proposed approach is only marginally better (1% or less).
- The discussion regarding the sub-optimal performance on DTD (LINES 216-221),  further highlights the niche application of the proposed approach. That being said, I wonder how/if the proposed approach would benefit dynamic textures (e.g. [a])? have the authors considered this?
- In terms of the method itself, the proposed approach should in fact replace GAP and I don't see a striking complementarity between the two. This comment is further confirmed in Table 2, which shows that GAP only yields a marginal contribution. Can the authors further comment on that?

[a] I. Hadji and R. P. Wildes,  "A New Large Scale Dynamic Texture Dataset
with Application to ConvNet Understanding", ECCV 2018

**Time Spent Reviewing:**

2

---

> ### Author Response · Authors · 2021-08-10
> **Responses to the comments from Reviewer H23L**
>
> We sincerely thank the reviewer for his/her constructive comments.
>
> -----------------------------------------------------------------------------
> **Regarding the performance of proposed approach**
>
> We would like to clarify that **the performance improvement of our method is not marginal**. Indeed, it is very difficult to have a large improvement over SOTA results on existing texture datasets, and the improvement brought by our method is not marginal but noticeable. To show this, we compare the relative improvement of DSRNet (CVPR-2020) over MAPNet (CVPR-2019) and the relative improvement of our method over DSRNet (CVPR-2020). Please refer to the table below for the results on four datasets. It can be seen that in 5/8 cases, the improvement of our method over DSRNet is noticeably larger than the improvement of DSRNet over MAPNet, and in 2/8 cases the improvement brought by our method is very close to that brought by DSRNet. Here we do not compare on the DTD dataset because the results of DSRNet and MAPNet seem to be obtained from their defined splits on the DTD dataset, as stated in their works, rather than the standard one provided by the DTD dataset itself. Since their data splits and codes are not publicly available, and they are quite complex to implement, we cannot compare DSRNet and MAPNet with the same splits. We will elaborate this more in revision.
>
> | Resnet-18 | KTH | FMD | GTOS | GTOS-M |
> | -------------- | :--------------: |  :--------------: |  :--------------: |  :--------------: |
> | DSRNet(CVPR-2020) – MAPNet(CVPR-2019) | 0.90 | 0.50 | 0.70 | 0.67 |
> | Ours – DSRNet (CVPR-2020) | 4.82 | 0.96 | 2.10 | 1.45 |
>
>
>
> | Resnet-50  | KTH | FMD | GTOS | GTOS-M |
> | -------------- |  :--------------: |  ------------------- |  :--------------: |  :--------------: |
> | DSRNet(CVPR-2020) – MAPNet(CVPR-2019) | 1.40 | 0.30/0.80 | 0.60 | 0.39 |
> | Ours – DSRNet(CVPR-2020) | 2.34 | 0.74 | 0.41 | -1.83 |
>
> * Note that the values 0.80/0.30 are computed based on the results of MAPNet from [27] and [28] respectively.
>
> -----------------------------------------------------------------------------
> **Regarding "the proposed solution is not broadly applicable"**
>
> Our work focuses on texture description, one fundamental problem in pattern recognition. Though it is probably inapplicable to the applications not related to texture, we would like to mention that the texture-related applications are very wide, as texture is one fundamental visual cue. In addition, our work provides a new CNN module that is capable of encoding the spatial distribution of convolutional features, which provides additional discrimination to histogram statistics that only considers numerical distribution rather than spatial distribution. In other words, in the applications where histogram-based statistics is used for encoding the numerical distribution of spatial elements, the FE module can be used to provide a different statistical measure on spatial distributive features.
>
> ------------------------------------------------------------------------------
> **Regarding the benefits of proposed method on dynamic textures**
>
> Thanks for the reference which will be added for discussion. Dynamic textures are the textures in motion, and one related focus is how to characterize the texture in both space and time domains. A powerful static texture descriptor can undoubtedly benefit the characterization on spatial texture feature of dynamic textures. In fact, many existing dynamic texture descriptors are extended from static textures; see e.g.
>
> "Dynamic Texture Recognition Using Local Binary Patterns with an Application to Facial Expressions, TPAMI 2007."
>
> "Spatiotemporal Lacunarity Spectrum for Dynamic Texture Classification. CVIU 2017."
>
> In addition, our FE module provides a deep learnable statistical method for describing the spatial distribution of spatial features. For dynamic textures, the characteristic motion patterns of textons will lead to specific spatial-temporal distributions of textons, which are quite important for the recognition. Thus, we believe the proposed approach could also benefit dynamic texture recognition. Since the extension of a static texture descriptor to a dynamic texture descriptor is nontrivial, we will leave it in our future work.
>
> ------------------------------------------------------------------------------
> **Regarding the complementarity between GAP and FE Module**
>
> The FE module generates the description by examining the variations of feature maps across scales, which can be roughly viewed as ‘high-pass’, while GAP characterizes feature maps via average, which can be viewed as ‘low-pass’. Thus, we think they can provide some complementary information. It can be seen from Table 2 that GAP does bring certain improvement (e.g., 1.3% on GTOS-Mobile), though the improvement varies over datasets. We would again clarify that such an improvement over SOTA results is nontrivial on existing texture datasets.
>
> In addition, our main contribution is proposing the FE module and FENet. We think combining with GAP does not lower the contribution of our work. Indeed, many existing approaches also contain GAP as a basic element; see e.g. [7.10,28].
>
> ------------------------------------------------------------------------------
> **Regarding "While the authors provided different ablations, they did not explicitly discuss the limitations of their approach. It would be useful to dedicate a small subsection to this as this would benefit progress on this domain."**
>
> We did provide a section in supplementary materials for studying the limitations of our method. Please refer to Section I (Study on Confusion Cases) in the supplementary materials. We will add more discussion and more failure cases in revision. Another limitation of our approach is that it is not suitable for contour-based or color-based recognition, as it mainly considers texture cues instead of color/contour cues. We will also add such a discussion in revision.

---

> > ### Comment · Reviewer_H23L · 2021-08-23
> > **reply to rebuttal**
> >
> > Thanks for addressing my comments.
> >
> > ## regarding performance
> > My comment on performance pertained to gaps of ~1% or less, which I continue to believe are *marginal*. The fact that previous work (i.e. DSRNet(CVPR-2020) vs. MAPNet(CVPR-2019) ), had such marginal improvements speaks for the sadly unfortunate state of computer vision rather than for significance of their results. For that reason, I would rather argue in favor of the new outlook of your method  on the problem rather than the corresponding results.
> >
> > ## Regarding the comment on broad applicability of the method
> > I agree that texture is an important problem in and of itself, however, the proposed method does not seem to apply across texture datasets (e.g. DTD), is the argument that texture in DTD are not *proper* textures?

---

> > > ### Author Response · Authors · 2021-08-24
> > > **Reply to reviewer's comments at the 2nd round**
> > >
> > > Thanks for the reviewer's comments.
> > >
> > > **Regarding "I agree that texture is an important problem in and of itself, however, the proposed method does not seem to apply across texture datasets (e.g. DTD), is the argument that texture in DTD are not proper textures?"**
> > >
> > > As discussed in the previous rebuttal, DSRNet and MAPNet seem to use different training/test splits from the standard ones (as stated in their works), which is probably one reason for the sub-optimal performance of our method to these two methods. Since their codes and splits are not available, we cannot compare with these two methods in the exactly-same setting. Let us compare our method with the ones that use standard splits and have public codes, say DeepTEN, DEPNet and HistNet, our method still shows superior performance. We will clarify it in revision.
> > >
> > > The discussion regarding the sub-optimal performance on DTD in our paper (LINES 216-221) is somehow misleading, and we will revise it. Here we would like to clarify that it does not imply the textures in DTD are not proper textures. Indeed, DTD is a novel and challenging benchmark dataset in the field of texture analysis. As stated in [a], the labels for the textures in DTD are based on describable subjective semantics, which are quite different from other texture datasets whose labels are based on objective materials. Even in such a case, our method still applies and provides a good texture descriptor (e.g. superior performance to DeepTEN, DEPNet and HistNet).
> > >
> > > If we forget about the possible performance gap brought by using non-standard training/test splits on DTD, then we guess one possible advantage of MAPNet over our method on DTD could be as follows. MAPNet is developed based on multiple attribute perception (MAP) which is more suitable for the subjective attributes on DTD (similar reason for DSRNet), while the spatial regularity captured by our method is more suitable for encoding the characteristics of objective materials (e.g. the superior performance of our method to MAPNet and DSRNet on other datasets).
> > >
> > > [a] Describing Textures in the Wild. CVPR 2014.
> > >
> > >
> > >
> > > **Regarding the performance**
> > >
> > > We agree with the reviewer that the performance pertained to gaps of 1% or less is not a significant result and the performance of existing methods seems to saturate on the datasets. If we check the results of ResNet-18 in the table at the previous rebuttal, there are three datasets where our method has accuracy improvement larger than 1% and even over 4.5%, and on the remaining dataset the improvement is nearly 1%.
> > >
> > > We also agree that for the reason of performance saturation of existing methods, a new outlook on the problem is more important than the results. We sincerely thank the reviewer's feedback on this, and wish our points in this round and the ideas provided in our work can be considered for the judgement of our work.

---

> > > > ### Comment · Reviewer_H23L · 2021-09-01
> > > > **Final Comment**
> > > >
> > > > * I thank the reviewers for acknowledging my concerns over significance of results and hope the comments about results will be taken into account in the revised manuscript.
> > > >
> > > > * Also thank you for clarifying the situation with respect to DTD.
> > > >
> > > > Taking the rebuttal and other reviews into account, I decided to increase my rating accordingly.

---

> > > > > ### Author Response · Authors · 2021-09-04
> > > > > **Thanks**
> > > > >
> > > > > Thanks for the reviewer taking account into our responses. We will revise the paper as suggested.

---

### Decision · Program_Chairs · 2021-09-27

**Decision:**

Accept (Poster)

**Comment:**

The paper addresses the important problem of texture representation in neural network models. The approach is interesting and well received by the reviewers, and I personally find that many industrial applications could benefit from it.